# Identification of Protosappanoside D from *Caesalpinia decapetala* and Evaluation of Its Pharmacokinetic, Metabolism and Pharmacological Activity

**DOI:** 10.3390/molecules27186090

**Published:** 2022-09-18

**Authors:** Yueting Li, Wensha Meng, Li Yuan, Li Jiang, Zuying Zhou, Mingyan Chi, Zipeng Gong, Xue Ma, Yong Huang, Lin Zheng

**Affiliations:** 1State Key Laboratory of Functions and Applications of Medicinal Plants, Guizhou Provincial Key Laboratory of Pharmaceutics, Guizhou Medical University, Guiyang 550004, China; 2School of Pharmacy, Guizhou Medical University, Guiyang 550004, China; 3Engineering Research Center for the Development and Application of Ethnic Medicine and TCM (Ministry of Education), Guizhou Medical University, Guiyang 550004, China

**Keywords:** Protosappanoside D, structure elucidation, metabolites, pharmacokinetics, anti-inflammation

## Abstract

Protosappanoside D (PTD) is a new component isolated from the extract of *Caesalpinia decapetala* for the first time. Its structure was identified as protosappanin B-3-*O*-*β*-D-glucoside by ^1^H-NMR, ^13^C-NMR, 2D-NMR and MS techniques. To date, the pharmacological activities, metabolism or pharmacokinetics of PTD has not been reported. Therefore, this research to study the anti-inflammatory activity of PTD was investigated via the LPS-induced RAW264.7 cells model. At the same time, we also used the UHPLC/Q Exactive Plus MS and UPLC-MS/MS methods to study the metabolites and pharmacokinetics of PTD, to calculate its bioavailability for the first time. The results showed that PTD could downregulate secretion of the pro-inflammatory cytokines. In the metabolic study, four metabolites were identified, and the primary degradative pathways in vivo involved the desaturation, oxidation, methylation, alkylation, dehydration, degradation and desugarization. In the pharmacokinetic study, PTD and its main metabolite protosappanin B (PTB) were measured after oral and intravenous administration. After oral administration of PTD, its T_max_ was 0.49 h, t_1/2z_ and MRT_(0–t)_ were 3.47 ± 0.78 h and 3.06 ± 0.63 h, respectively. It shows that PTD was quickly absorbed into plasma and it may be eliminated quickly in the body, and its bioavailability is about 0.65%.

## 1. Introduction

*Caesalpinia decapeta* (*C. decapetala*) (Latin name: *Caesalpinia decapetala* (Roth) Alston) is a traditional medicinal plant of the Miao people. It belongs in the Fabaceae family and has been used to treat colds, fever, rheumatic pain, etc. for many years [1,2]. In previous studies, it was found that the extract of *C. decapetala* mainly contains compounds such as phenolics, flavonoids and terpenoids, which has good anti-inflammatory, antiviral and antitumor properties [3,4,5]. For example, gallic acid has a good proapoptotic effect on A-489 renal cancer cells [6], and can also inhibit the secretion of inflammatory factors induced by LPS [7]; sappanchalcones can significantly reduce the secretion of tumor necrosis factor alpha (TNF-α), interleukin-1β (IL-1β) and interleukin-6 (IL-6) to improve the joint destruction and surface erosion claws of CIA mice [8,9]; luteolin can play an anti-inflammatory role by inhibiting NF-κB pathway [10], and inhibiting the STAT3 pathway, showing a strong ability to inhibit liver fibrosis [11]; protosappanin B (PTB) has obvious pharmacological activities such as antitumor, antioxidant and anti-inflammatory activities [12,13]. Protosappanoside D (PTD), a novel dibenzoxocin derivative, has been discovered in the extract of *C. decapetala* by the ^1^H-NMR, ^13^C-NMR, 2D-NMR and MS methods for the first time. The structure of PTD was very similar to that of PTB. Therefore, it is speculated that PTD may have good activity. Further studies on PTD are warranted.

Pharmacokinetic studies of the absorption, distribution, metabolism and excretion of drugs in the body are important to the research and development of new drugs [14]. In some cases, natural products may act as prodrugs, they are converted by metabolic activation into an active pharmacologic agent, which exerts pharmacological effects [15]. For example, baicalin and baicalein are the main chemical components of Scutellaria baicalensis (Georgi) [16], baicalin may undergo removal of its glycosidic bonds and thus be converted into baicalein, which has good pharmacological activity and bioavailability through enzyme-catalyzed reactions and ultrasonic extraction in previous studies [17,18]. After identification, PTD can be transformed into PTB after the removal of its glycosidic bonds, and PTB has obvious pharmacological activities. Therefore, it is important to study the metabolites and processes of PTD in vivo.

An effective analytical method must be utilized to analyze the complex chemical mixtures found in biological samples. Therefore, in the present study, ultra-high performance liquid chromatography-quadrupole/electrostatic field orbital trap high resolution mass spectrometry (UHPLC/Q Exactive Plus MS) technology was implemented to study the metabolism of PTD, and the ultra-performance liquid chromatography-tandem mass spectrometry (UPLC-MS/MS) method was used for the quantitative study of PTD in rat blood plasma [19,20,21]. At the same time, we also used an in vitro model of inflammation using lipopolysaccharide (LPS) induced inflammatory response in RAW264.7 cells to study the anti-inflammatory activity of PTD. Through the above experiments, this study will report the pharmacological activity, pharmacokinetics and metabolism of PTD for the first time, so as to provide a scientific basis for in-depth research and drug development of PTD.

## 2. Results

### 2.1. Structural Qualification of PTD

PTD was isolated as white solid. The molecular formula was established by HR-QTOF-MS, ^1^H-NMR and ^13^C-NMR spectra data (*m*/*z*: 465.1385 [M-H]^−^, C_22_H_26_O_11_, calculated for 465.1402). According to the ^1^H-NMR and ^13^C-NMR spectra data of PTD, it was found that it is basically consistent with the mixture of protosapanoside A and isoprotosapanoside A was reported in the literature [22]. However, the positions of sugar substituents in the compound are different, and the compound is substituted at carbon 3, so the compound is also a mixture of two dibenzoxyoctyl glycosides.

^1^H-NMR spectrum (600 MHz, CD_3_OD) (Figure 1A) showed the signals for two 1,2,4-trisubstituted benzene ring systems in the low field region: δ_H_ 7.10, 7.10 (1H, 1H, d, *J* = 8.5 Hz, H-1), 6.89, 6.84 (1H, 1H, d, *J* = 8.5 Hz, H-2 ), 6.84, 6.78 (1H, 1H, d, *J* = 2.5 Hz, H-4); two 1,2,4,5-tetrasubstituted benzene ring systems: δ_H_ 6.74, 6.69 (1H, 1H, s, H-12), 6.75, 6.75 (1H, 1H, m, H-9); two sugar terminal stromal signals δ_H_ 4.96, 4.94 (1H, 1H, D, *J* = 8.1, 7.6 Hz, Glc-1′), according to its coupling constant, it could be inferred that the glycoside bond was *β* configuration; eight oxymethylene hydrogen signals δ_H_ 4.42, 4.19 (1H, 1H, m, H-6b), 3.92, 3.58 (1H, 1H, d, *J* = 11.8, 12.3 Hz, H-6a), 3.56, 3.41 (1H, 1H, overlapped, C7-CHa), 3.56, 3.49 (1H, 1H, overlapped, C7-CHb); two groups of hydrogen signals coupled with carbon δ_H_ 2.72, 2.67 (1H, 1H, d, *J* = 13.6 Hz, H-8CHb), 2.56, 2.52 (1H, 1H, d, *J* = 13.6 Hz, H-8CHa). ^13^C-NMR (150 MHz, CD_3_OD) (Figure 1B) spectrum depicted 44 carbon signals, including two glucopyranose signals (δ_C_ 102.2, 102.2, 78.1, 78.1, 78.0, 78.0, 74.9, 74.9, 71.4, 71.4, 62.5, 62.5); six methylene carbon signals (δ_C_ 76.9, 76.8, 65.8, 68.4, 42.6, 40.0); two seasonal carbon signals (δ_C_ 73.2, 72.9). The other signals are the carbon signals on the benzene ring (δ_C_ 160.4, 159.5, 159.3, 159.3, 145.2, 145.2, 145.1, 145.1, 132.8, 132.1, 132.1, 131.6, 128.3, 127.9, 127.6, 127.0, 120.1, 119.1, 117.9, 117.6, 113.1, 112.6, 110.6, 110.0). According to the HMBC spectrum (Figure 1C), H-1 is related to C-3, H-2 is related to C-4/C-12b, H-4 is related to C-12b/C-3/C-4a, H-6 is related to C-4a/C-7/C-8, H-8 is related to C-6/C-7/C-7a/C-9/C-8a/C-12a, H-9 is related to C-8/C-11, H-12 is related to C-10/C-8a/C-12b, and H-1′ is related to C-3, indicating that the glucose group is connected to C-3.

All hydrocarbon signals of PTD were accurately assigned by integrating MS and NMR data (Table 1). After searching, PTD was identified as a pair of new compounds, and its chemical structure was protosappanin B-3-*O*-*β*-D-glucoside, named protosapanoside D (1a) and isoprotosapanoside D (1b) as shown in Figure 2.

### 2.2. In Vitro Pharmacodynamic Experiment

#### 2.2.1. Effects of Tested Drugs on RAW264.7 Cells Viability

The results of MTS assay (Figure 3) showed that there was no cytotoxicity to RAW264.7 cells by PTD treatment below 200 μmol/L and PTB treatment below 300 μmol/L. Subsequent experiments were conducted with 200 μmol/L PTD and PTB.

#### 2.2.2. Effects of PTD and PTB on the Secretion of IL-6, TNF-α, IL-1β, and NO by RAW264.7 Cells

Many proinflammatory factors are secreted during inflammation, but excessive secretion of inflammatory factors will aggravate inflammation, so inhibiting the secretion of inflammatory factors is of great significance for the treatment of inflammation. The results are shown in Figure 4, the release of IL-6, TNF-α, IL-1β and NO in the supernatant of RAW264.7 cells were significantly induced by LPS. Compared to the model group, PTD and PTB significantly reduced the secretion of TNF-α, IL-1β and IL-6, and the efficacy of the two drugs was similar to that of dexamethasone, which proves that they have good anti-inflammatory effects.

### 2.3. Metabolites Identification

#### 2.3.1. Mass Spectrometric Analysis of PTD

To identify the metabolites, a comprehensive understanding of the characteristic product ions and fragmentation pathways of the parent compounds is necessary. In the present study, chromatographic and mass spectrometry conditions were optimized with PTD.

PTD showed a HPLC profile with a retention time of 4.32 min. In negative scan mode (Figure 5A), a peak of the parent compound PTD, [M-H]^−^ at *m*/*z* 465.1402 (C_22_H_26_O_11_), was obtained along with a high collision energy spectrum of PTD and a series of peaks for important fragment ions. PTD was detected as a deprotonated [M-H]^−^ion at *m*/*z* 465.1400 (C_22_H_2__6_O_11_). After removal of glucoside from PTD, an ion is generated at *m*/*z* 303.0871(C_16_H_16_O_6_^−^). This ion after the prominent loss of H_2_O and CH_2_O, mass fragmentation occurred on the side-chain radical and eight-element ring skeleton. The ion at *m*/*z* 285.0767 (C_16_H_13_O_5_^−^, due to loss of H_2_O) formed a double bond. The ion at *m*/*z* 303.0871 further lost a neutral fragment of CH_2_O, which opened the eight-element ring, generating an ion at *m*/*z* 273.0768 (C_15_H_13_O_5_^−^). Then this ion further lost a molecule of H_2_O to generate the ion at *m*/*z* 255.0660 (C_15_H_11_O_4_^−^) and lost a molecule of C_2_H_3_O to generate the ion at *m*/*z* 243.0659 (C_14_H_11_O_4_^−^). The ion at *m*/*z* 273.0768 (C_15_H_13_O_5_^−^) lost a neutral fragment of C_2_H_2_O, generating the ion at *m*/*z* 231.0758 (C_13_H_11_O_4_^−^), and further lost an atom of O to generate the ion at *m*/*z* 215.0704 (C_13_H_11_O_3_^−^). The proposed mechanism for fragment ion generation was shown in Figure 5B. The fingerprint product ions could serve well in the screening and identification of metabolites derived from PTD [23].

#### 2.3.2. Metabolites Identification

Metabolites were detected and identified in blood plasma samples. Full scans and product ion scans in positive ion mode were conducted for all samples. As Appendix A shows, four metabolites were identified by comparison between blank and administered samples. The metabolic messages of PTD are shown in Table 2 and the predicted metabolic pathways are depicted in Figure 6. It was found that the deprotonated molecule [M-H]^−^ ion at *m*/*z* 303.0874 was PTB by comparing the chromatograms of the reference standards, blank plasma, rat administered plasma and deducted blank plasma (Figure 7), and its response was high. At present, studies have reported that PTB has antitumor and anti-inflammatory activities, so it was necessary to study the pharmacokinetics of PTD and its metabolite PTB.

### 2.4. Pharmacokinetic Study

#### 2.4.1. UPLC-MS/MS Method Validation

##### Specificity

The MRM chromatograms of blank plasma, blank plasma spiked with standard solutions and IS, and a pharmacokinetic sample at 30 min after oral PTD administration are shown in Figure 8. The retention times of PTD, PTB, and IS were 0.74, 1.46 and 1.42 min, respectively. No endogenous interference was observed in MRM mode for any analyte.

##### Linearity and Lower Limit of Quantification

Typical calibration curves, linearity ranges, coefficients of correlation, and LLOQ for PTD and PTB are shown in Table 3. The two analytes exhibited good linearity, with correlation coefficients varying from 0.9958–0.9998.

##### Precision and Accuracy

The results for intra-day and inter-day precision and accuracy for QC samples at three concentrations are shown in Table 4. The results showed that the RSD of intra- and inter-day measurements was less than 15%, and the accuracy range was approximately ranging from 86.06–109.38%. These results indicate that the proposed method is precise and accurate.

##### Extraction Recovery and Matrix Effect

The results of the matrix effect and extraction recovery of PTD and PTB are summarized in Table 5. The extraction recoveries of the two analytes were within the range of 87.65–103.25%, indicating that the recovery of analytes was consistent and reproducible. The matrix effects ranged from 88.83–105.08%, suggesting an insignificant matrix effect on the quantification of PTD and PTB.

##### Stability

The sample stability results (*n* = 5) are summarized in Table 6. The results showed that all three analytes were stable under all testing conditions, including at room temperature for 24 h, refrigeration at 4 °C for 24 h, and three freeze-thaw cycles. The RSD of all the samples under the above conditions were both within 15%.

#### 2.4.2. Pharmacokinetic Study

The method established above was applied to determine the pharmacokinetics of PTD in rats. After oral administration (27.3 mg/kg) and intravenous administration (9.1 mg/kg) of PTD, the plasma concentration versus time profiles were shown in Figure 9 and Figure 10, and the pharmacokinetic parameters were listed in Table 7. Both administration methods could detect PTD and its metabolite PTB. PTD reached a peak concentration rapidly at about 0.49 h after oral administration. The t_1/2z_ of oral and intravenous administration were 3.47 ± 0.78 h and 3.65 ± 1.26 h, respectively, whereas MRT_(0–t)_ was 3.06 ± 0.63 h and 1.80 ± 0.50 h, respectively. The result suggested that it has a fast absorption rate with a short residence time. Thus, it can be quickly absorbed into blood circulation and eliminated rapidly from the body. The studies had shown that the exposure of PTD in vivo after intravenous injection is significantly higher than the exposure after oral absorption, and its bioavailability was about 0.65%.

## 3. Discussion

PTD was identified in the extract of *C. decapetala* for the first time. Through NMR and MS identification methodology, the chemical structure of the mother nucleus was found to be very similar to that of PTB (refer Figure 2), of which are the tricyclic mother nuclei of dibenzo oxygen-containing eight-membered heterocycles. According to prior studies, PTB has a good anti-inflammatory effect [24]. Based on structure-function similarities, we speculated that PTD might also have good anti-inflammatory activity.

Based on these similarities and assumptions, we chose the in vitro inflammatory model utilizing RAW264.7 cells, induced by LPS to detect the efficacy of PTD and PTB as anti-inflammatory agents [25,26]. The experimental results showed that PTD and PTB both significantly downregulated the secretion of TNF-α, IL-1β, and IL-6 by the RAW264.7 cells, compared to the model group, and the anti-inflammatory effects of PTB were similar to those previously reported. A comparison of the anti-inflammatory activities of PTD and PTB showed similar activity for both compounds.

Due to the high similarity between the parent heterocyclic nuclei of PTD and PTB, it was speculated that they may be prototypes or metabolites of each other. Prior studies had found that PTB and PTD in the extract of *C. decapetala* were able to enter the bloodstream. To eliminate any interference of PTB in the extract, PTD was administered orally and intravenously to explore the dynamic and transformation processes of PTD in vivo.

Studies of the degradative metabolism of these compounds proved that the primary metabolic pathways in vivo involved the organic chemical desaturation, oxidation, methylation, alkylation, dehydration, degradation, and desugarization. Previous studies showed that the in vivo efficacy of the bioactive components depends on their blood concentration and the activity of their metabolites. Therefore, it is of great significance to study the metabolites of PTD. Preliminary experiments were performed to compare the in vivo pharmacokinetic processes of these metabolites. The results showed that PTB was the main active metabolite of PTD, not only because its content was high, but also because only PTB had a complete in vivo kinetic process. PTB is a metabolite of PTD, which has great utilization value because of its anti-inflammatory, anti-cancer, antioxidant and other activities. Thus, PTD can be used as a prodrug to exploit its value through structural modification and transformation [27,28].

We further investigated the kinetics of PTD and its main metabolite PTB in vivo. A rapid, reliable and sensitive UPLC-MS/MS method was established and validated for simultaneous determination of PTD and its main metabolite PTB in rat plasma for the first time in pharmacokinetic study. The positive and negative modes were scanned simultaneously, and the scanning mode was selective ion monitoring (MRM) and the analysis time was 3 min, which considerably improved the experimental efficiency. The results of the experimental methodology met the requirements for its validation for biological sample analysis and determination, and were simple, selective and sensitive. This method was successfully used to study the pharmacokinetic processes of PTD and PTB, and relevant pharmacokinetic parameters were obtained. The experimental results showed that the T_max_, t_1/2z_, and MRT_(0–t)_ of PTD administered orally in rats were 0.49 h, 3.47 h and 3.06 h, respectively, indicating that PTD can be absorbed into the blood circulation rapidly and transformed rapidly into PTB and other metabolites with a short residence time. Bioavailability is a measure of drug absorption in organisms, where the bioavailability of drugs is higher. The results showed that the AUC_(0–t)_ of PTD after intravenous administration was significantly higher than that oral administration. Its absolute bioavailability was low (approximately 0.65%), which indicates that the total amount of PTD entering the systemic circulation was significantly reduced after oral administration. It is speculated that some drugs are not absorbed due to the first-pass effect, resulting in their low absolute bioavailability. However, if a drug has low bioavailability but high therapeutic exposure, it may also have high developmental significance for future drug development, and other methods should be used to modify or transform them to make full use of their value. Based on formula, CL = kV, t_1/2_ = 0.693/k. The t_1/2_ of PTD after oral and intravenous injection is close, indicating that k is close after two kinds of administration, which proves that CL_z/F_ is mainly affected by V_z/F_. The CL_z/F_ and V_z/F_ after oral PTD were much greater than those after intravenous injection. But that V = X0/C (X0: in vivo dose; C: plasma concentration). Therefore, we speculated that after oral PTD, intestinal flora or some metabolic enzymes make PTD metabolize rapidly in the gastrointestinal tract, resulting in low plasma drug concentration of PTD. Thus, the CL_z/F_ and V_z/F_ of oral administration were higher than those of intravenous administration. After oral administration, the AUC_(0–t)_ of PTD was 382.17 ng/mL·h, and the AUC_(0–t)_ of PTB was 179.33 ng/mL·h. The PTB content in vivo was approximately half of that of PTD, indicating that PTD was mainly transformed into PTB. It was speculated that the efficacy of PTD in vivo might be attributed to the role of PTB and other metabolites. This hypothesis needs to be verified through further experiments [29,30,31].

This study had provided the structural identification, determined the anti-inflammatory efficacy, and in vivo pharmacokinetic processes of PTD, which can be used as the basic research before clinical application of PTD and is of great significance to the development and utilization of PTD as a therapeutic agent.

## 4. Materials and Methods

### 4.1. Drug and Reagents

*Caesalpinia decapeta* was purchased from the medicinal materials market Wandongqiao, Guiyang City, Guizhou Province, China (2019 harvest), and it was identified as the dried root of *Caesalpinia decapetala* (Roxb.) Alston by Associate Professor Chunhua Liu (School of Pharmacy, Guizhou Medical University). Protosappanoside D (PTD, purity ≥ 99%, it was isolated from the extract of *Caesalpinia decapeta*, and its extraction and separation data are shown in the results Section 2.1), protosappanin B (PTB, purity ≥ 98%; Chengdu AIFA Biotechnology Co., Ltd., Chengdu, China) and puerarin (internal standard, IS) standards were purchased from the China Food and Drug Administration Research Institute; lipopolysaccharide (LPS) and dexamethasone (Dex) were obtained from Sigma-Aldrich (St Louis, MO, USA), and the nitric oxide assay kit was obtained from Nanjing Jiancheng Biotechnology Co., Ltd. (Nanjing, China). Tumor necrosis factor alpha (TNF-α), interleukin-1β (IL-1β) and interleukin-6 (IL-6) ELISA kits were purchased from Proteintech Group Inc. HPLC-grade acetonitrile, methanol and formic acid were obtained from Merck (Darmstadt, Germany). Distilled water was supplied by Watsons Group Co. Ltd. (Hong Kong, China). All other solvents used in the present study were of analytical grade and commercially available.

### 4.2. Extraction and Separation of PTD

The crude powder of *Caesalpinia*
*decapeta* was weighed at 15 kg, which was then cold soaked and extracted with 8 volumes of 80% ethanol for 5 d, filtered to obtain the extract, and concentrated to 20 L. This was separated with D101-8 macroporous resin, eluted with 70 L of water, 20 L of 50% ethanol and 30 L of 90% ethanol; further, 50% ethanol eluent was collected and concentrated to 2.5 L. The concentrated solution was again separated with D101-8 macroporous resin and washed, in which 90 L with water, 25 L with 50% ethanol, and 15 L with 20% ethanol. Then, 50% ethanol eluent was collected and concentrated to 250 mL. Purification at medium pressure and gradient elution with water, 5% acetonitrile, 10% acetonitrile, 25% acetonitrile, 35% acetonitrile, 50% acetonitrile and 90% acetonitrile were conducted; further, 25% acetonitrile were collected and concentrated to a small volume. We added 50% methanol water to 100 mL, and the Welch Uitimate XB-C_18_ column (50 × 250 mm, 10 μm) was washed with 7% acetonitrile water at a flow rate of 70 mL/min for high-pressure preparation and collection of the target stream. The organic solvent was concentrated at 50 °C and lyophilised to obtain a 1.0 g sample. This was dissolved in 30 mL pure water. A Kromasil C4 column (30 × 250 mm, 5 μm) was used. The extract was washed with 9% acetonitrile water at a flow rate of 40 mL/min for secondary high-pressure preparation, and the target stream was collected. The organic solvent was concentrated at 50 °C and lyophilised to obtain PTD.

### 4.3. Pharmacodynamic Experiment

#### 4.3.1. Cell Culture and Viability Assessment

RAW264.7 cells, obtained from American Type Culture Collection (ATCC, Manassas, VA, USA), were cultured in Dulbecco’s modified Eagle’s medium (DMEM) (Life Technologies, Grand Island, NY, USA) supplemented with 10% fetal bovine serum (FBS) and 1% penicillin-streptomycin (Grand Island, NY, USA). The cultures were incubated at 37 °C in a humidified atmosphere containing 5% CO_2_.

Cell viability was determined using the MTS cell viability assay kit. RAW264.7 cells were cultured at a density of 1 × 10^5^ cells per well in 96-well plates by the treatment with various concentrations of PTD and PTB, respectively. After 24 h, 5 µL MTS was added to each well and incubated for 3 h. The plates were then read at 490 nm using a microplate reader [32].

#### 4.3.2. Measurement of Cytokines and Nitrite Levels

RAW264.7 cells (1 × 10^5^ cells/well in a 96-well plate) were treated with PTD, PTB (200 μmol/L) and stimulated by LPS (1 μg/mL) for 24 h. As a positive control, the impacts of Dex on LPS-induced RAW264.7 cells were also determined parallelly. The supernatants were then collected, and the accumulated NO level was determined using Griess reagent. The concentrations of TNF-α, IL-6 and IL-1β in conditioned media were measured using enzyme-linked immunosorbent assay (ELISA) according to the supplier’s instructions.

### 4.4. Animal Studies

Specific pathogen-free Sprague-Dawley rats (230 ± 10 g) were purchased from Changsha Tianqin Biotechnology Co., Ltd. (Changsha, China; certificate no. SCXK (Xiang) 2014-0010). Before starting the experiments, the rats were allowed one week of acclimatization in the animal quarters under air conditioning (25 ± 1 °C) and an automatically controlled photoperiod of 12 h of light daily. They were fed with standard rodent chow and tap water ad libitum. Rats were fasted for 12 h before the start of the experiment but were free to drink water. All experiments were approved by the Animal Ethics Committee of Guizhou Medical University and conducted in accordance with the guidelines of the Committee on the Care and Use of Laboratory Animals in China.

To investigate the PTD metabolites, Sprague-Dawley rats were randomly divided into two groups with six rats in each group. PTD was continuously administered by gavage twice daily for 3 d at a dose of 27.3 mg/kg. Blood was collected from the rat femoral artery 2 h after the last administration, centrifuged at 6000 rpm at 4 °C for 6 min, and 100 μL of plasma was separated and frozen at −20 °C until analysis.

For the pharmacokinetic study, 12 rats were randomly divided into two groups, with 6 rats in each group. In the oral administration group, PTD was intragastrically administered to the rats at a dose of 27.3 mg/kg. For intravenous administration, PTD was injected at a dose of 9.1 mg/kg via the tail vein. Blood samples (0.3 mL) were collected directly from the caudal vein using anticoagulant centrifugal tubes at 0.083, 0.167, 0.33, 0.5, 0.75, 1, 1.5, 2, 4, 6, 9, 12 and 24 h after intravenous and oral administration, respectively. The blood sample was centrifuged at 6000 rpm for 6 min, and the supernatant plasma was transferred into a new clean tube and stored at −80 °C prior to further use.

### 4.5. UHPLC/Q Exactive plus MS Conditions

Qualitative analysis of metabolites was conducted on a Vanquish horizon (Thermo Scientific, Waltham, MA, USA) and combined with a Q Exactive Plus Orbitrap MS (QE Plus, Thermo Scientific, USA) system. The separation of PTD and its metabolites was achieved on a Hypersil gold column (2.1 × 150 mm, i.d., 1.9 μm). The mobile phase consisted of water containing 0.1% formic acid (B) and acetonitrile containing 0.1% formic acid (A). The gradient elution program was set as follows: 0–2 min, 5% A; 2–3 min, 5~25% A; 3–6 min, 25~40% A; 6–13 min, 40~95% A; 13–14 min, 95~95% A; 14–15 min, 95~5% A; 15–18 min, 5~5% A; flow rate, 0.3 mL/min. The MS unit parameters were set as follows: the range of data acquisition was 150–1000 *m*/*z*, electrospray voltage was 2.5 kV(−), 3.5 kV(+), solvent removal gas flow rate was 8 L/min, and desolvent gas temperature was 320 °C. Acquisition and processing were conducted using the Xcalibur 4.2 software (Thermo Fisher Scientific, USA) and Compound Discoverer 3.2 (Thermo, USA).

### 4.6. UPLC-MS/MS Conditions

The UPLC-MS/MS system consisted of an ACQUITY I-Class UPLC system and an XEVO TQ-S triple-quadrupole tandem mass spectrometer (Waters Corp., Milford, MA, USA) equipped with an ESI source. Chromatographic separation was performed using a UPLC BEN C_18_ column (2.1 × 50 mm, 1.7 m; Waters Corp., Milford, MA, USA) at 40 °C. The mobile phase consisted of water containing 0.2% formic acid (B) and acetonitrile containing 0.2% formic acid (A) delivered at 0.3 mL/min as a gradient as follows: 0–0.5 min, 10% A; 0.5–3 min, 90% A; 3–4 min, 90% A; 4–4.5 min, 10% A; 4.5–5 min, 10% A. Waters XEVO TQ-S triple-quadrupole mass spectrometer equipped with an ESI source was used for the mass analysis and detection. All analytes were quantified in the MRM mode, and the optimised MS parameters are listed in Table 8. The main working parameters were set as follows: source temperature, 120 °C; desolvation temperature, 350 °C; and nebuliser gas (N_2_), 650 L/h. The scan time for all analytes was set at 0.2 s and acquisition and processing were conducted on a Micromass Masslynx4.1.

### 4.7. Sample Pretreatment

#### 4.7.1. Metabolite Research

A 600 μL aliquot of each plasma sample was transferred into a 5 mL centrifuge tube, 300 μL of 2% formic acid solution and 2.4 mL of methanol was added before vortexing for 2 min. This was followed by sonication in an ultrasonic machine for 10 min, and centrifugation at 13,225× *g* for 10 min. The supernatant was then placed in a centrifuge tube and evaporated to dryness under a stream of nitrogen at 37 °C. The residues were dissolved in 200 μL of 50% methanol and centrifuged at 13,225× *g* for 10 min, and 1 μL of the solution was injected into the QE system.

#### 4.7.2. Pharmacokinetic Study

A 100 μL aliquot of each plasma sample was transferred into a 1.5 mL centrifuge tube, and 50 μL of 2% formic acid solution was added before vortexing for 30 s. Subsequently, the internal standard (IS) (50 μL of 20 ng/mL puerarin) and 400 μL of methanol were added and vortexed for 1 min. For subsequent treatment methods were the same as mentioned in the above section (4.7.1); finally, 1 μL of the solution was injected into the UPLC-MS/MS system.

#### 4.7.3. Preparation of Standards and Quality Control (QC) Samples

Stock solutions of PTD, PTB and IS were separately weighed and dissolved in methanol to obtain final concentrations of 1.002, 1.004 and 0.495 mg/mL, respectively. Four sets of working standards were obtained by mixing and diluting the stock solutions in methanol. An appropriate amount of IS was dissolved in 50% methanol and diluted to 20 ng/mL. A mixture of working standard solutions (100 μL) and IS solutions (50 μL) was added to blank rat plasma (100 μL) to prepare the calibration standard solutions at final concentrations of 1–30,000, 1–1000 and 4 ng/mL for PTD, PTB and IS, respectively. QC plasma samples containing PTD (5.01, 250.5, 15,030 ng/mL) and PTB (2.51, 20.08, 251 ng/mL) were prepared in the same manner.

### 4.8. Method Validation

The analytical method was fully validated for specificity, precision, accuracy, extraction recovery, matrix effect, linearity, lower limit of quantification (LLOQ) and stability, according to the nonclinical drug pharmacokinetic study technical guidelines (China Food and Drug Administration 2014) and the Bioanalytical Method Validation Guideline (Chinese Pharmacopoeia 2015, Vol. 4).

#### 4.8.1. Specificity

The specificity of the method was assessed by comparing the chromatograms of blank plasma from two different rats and blank plasma spiked with PTD, PTB and the IS. This test determines where there is potential interference from endogenous substances with the analytes and the IS.

#### 4.8.2. Linearity and LLOQ

The linearity of the calibration curves for PTD and PTB was evaluated using nine appropriate concentrations and by plotting the area ratio of analytes to IS (y) versus the nominal concentration of analytes (x) using 1/x^2^ weighted least-squares linear regression. The LLOQ was the concentration of the lowest calibration standard with a signal-to-noise ratio that exceeded 10, along with an accuracy and precision of ±20%.

#### 4.8.3. Precision and Accuracy

Intra- and inter-day precision and accuracy were assessed by analyzing QC samples at low, medium and high concentrations, on three successive days. Precision was expressed as RSD.

#### 4.8.4. Extraction Recovery and Matrix Effect

The extraction recovery of PTD and PTB at low, medium and high QC levels was detected by comparing the peak areas of analytes obtained from rat plasma samples with the average peak areas of the same analytes spiked before and after extraction. Matrix effects were determined by comparing the peak areas of the analytes in the spiked post-extraction samples with those of the same analytes dissolved in 50% methanol at the same concentrations. Similarly, the extraction recovery and matrix effect of IS were determined. All QC sample analyses were performed in five replicates.

#### 4.8.5. Stability

The stability of analytes in rat plasma was assessed using QC samples at low, medium and high concentrations (*n* = 5 for each concentration level) under three conditions: (1) at room temperature for 24 h; (2) refrigerated at 4 °C for 24 h; and (3) three freeze-thaw cycles.

### 4.9. Data Analysis

Compound discoverer 3.2 was used for the analysis of metabolites of PTD; WinNonlin was used for the analysis of pharmacokinetic parameters, and Graphpad 8.2 was used for the illustrations. Data are expressed as mean ± SD. Statistical analysis was performed using the Statistical Product and Service Solutions software package (SPSS 22.0, SPSS Inc., Chicago, IL, USA). The results were analyzed using one-way ANOVA and bioavailability (F) was calculated according to the equation reported in He et al. [33].

## 5. Conclusions

PTD is a compound found in the extract of *C. decapetala*, which was isolated for the first time. Its structure was identified as protosappanin B-3-*O*-*β*-D-glucoside. In this study, we found that PTD could downregulate the secretion of TNF-α, IL-1β and IL-6, demonstrating its efficacy as an anti-inflammatory agent, and its efficacy is similar to that of PTB in anti-inflammatory potency. In the metabolic study, four metabolites were identified, and the primary degradative metabolic pathways in vivo involved desaturation, oxidation, methylation, alkylation, dehydration, degradation and desugarization. Pharmacokinetic studies showed that PTD was quickly absorbed and transformed rapidly in vivo, mainly into PTB, and its bioavailability in vivo was 0.65%. This was the first report of the pharmacological activity and pharmacokinetics of PTD, which will be of great significance for its development and utilization as a therapeutic agent in the future.

## Figures and Tables

**Figure 1 molecules-27-06090-f001:**
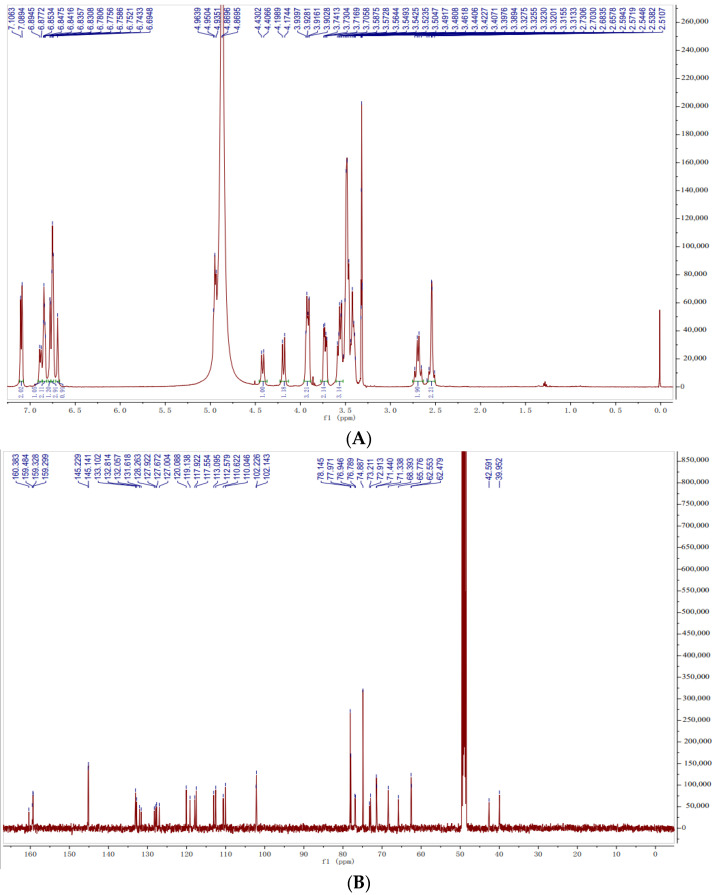
The hydrogen spectrum (**A**), carbon spectrum (**B**) and main HMBC correlations (**C**) of PTD.

**Figure 2 molecules-27-06090-f002:**
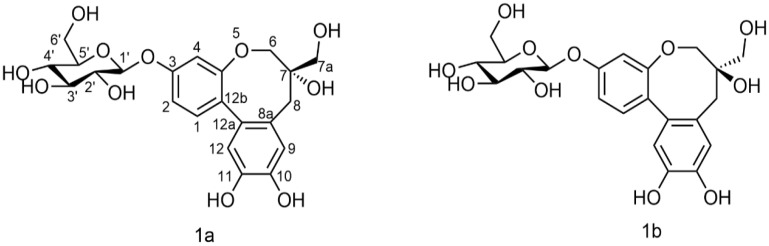
Structures of PTD: (**1a**) protosapanoside D, (**1b**) isoprotosapanoside D.

**Figure 3 molecules-27-06090-f003:**
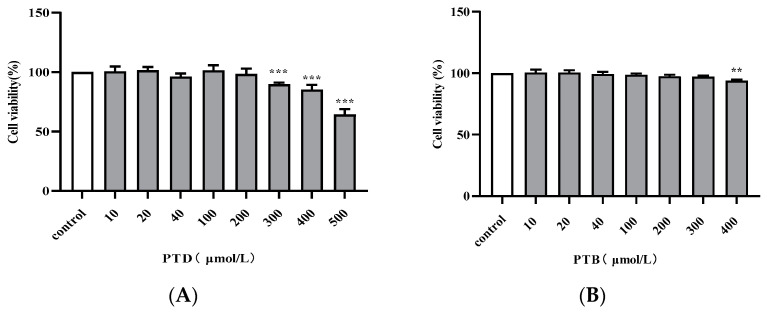
Cytotoxic effects of PTD (**A**) and PTB (**B**) on RAW264.7 cells. The values are expressed as the means ± SD (** *p* < 0.01, *** *p* < 0.001, compared to control group).

**Figure 4 molecules-27-06090-f004:**
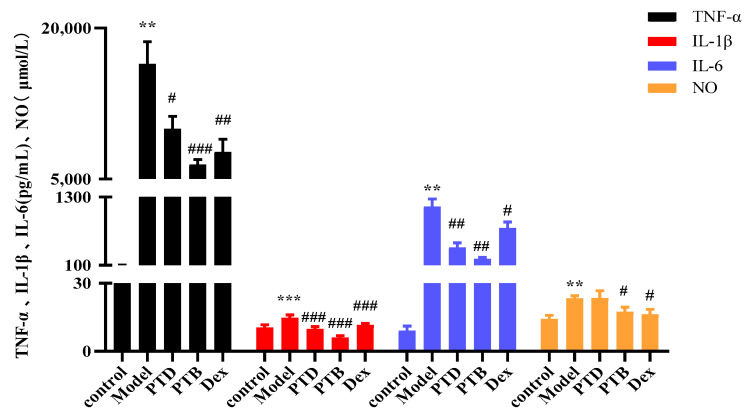
Effects of PTD on the release of TNF-α, IL-1β, IL-6 and NO in LPS-stimulated RAW264.7 cells. Dex was chosen as a reference control. The values are expressed as the means ± SD (** *p* <0.001, *** *p* < 0.001, comparison between control group and model group); (# *p* < 0.05, ## *p* < 0.01, ### *p* < 0.001, comparison between treated group and model group).

**Figure 5 molecules-27-06090-f005:**
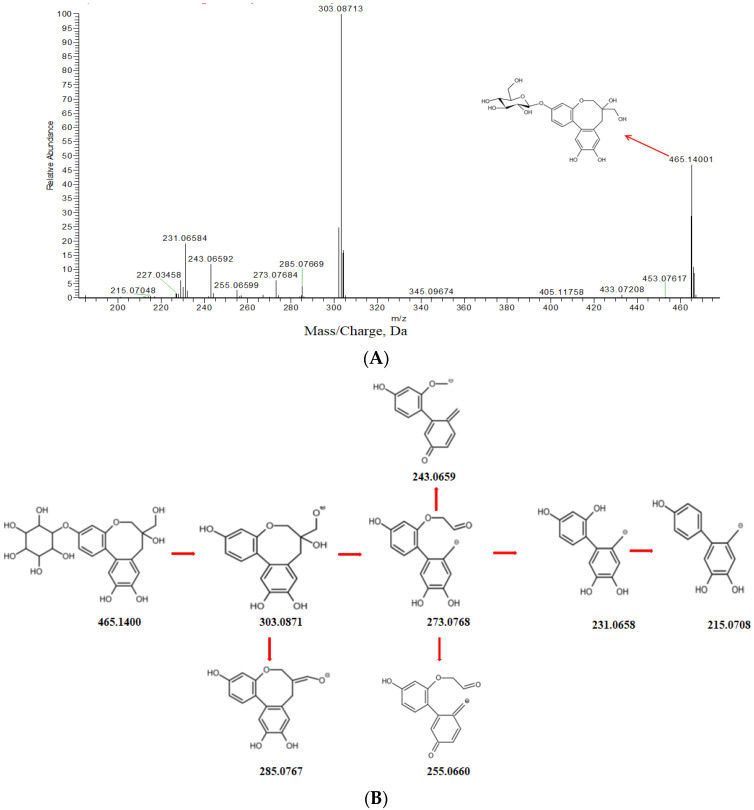
UHPLC/Q Exactive Plus MS/MS spectrum of PTD (**A**) and its proposed fragmentation pattern (**B**) in negative ion mode.

**Figure 6 molecules-27-06090-f006:**
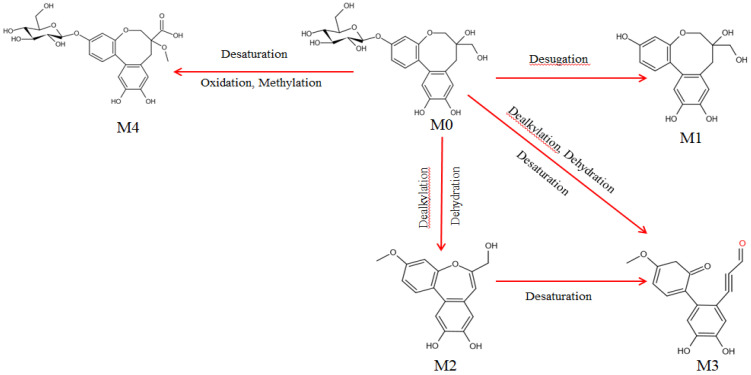
The proposed metabolic pathways of PTD in plasma.

**Figure 7 molecules-27-06090-f007:**
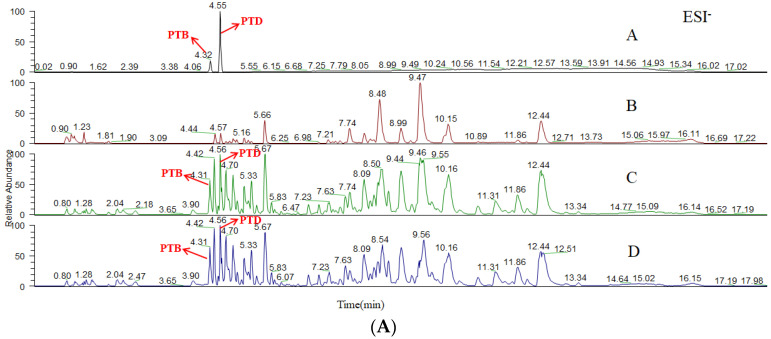
The negative base peak chromatograms (**A**) and the positive base peak chromatograms (**B**) of the biosamples of rats before and after intragastric administration of PTD. A, B, C and D represent chromatograms of chemical reference standards, blank plasma, administered plasma and deducted blank plasma, respectively.

**Figure 8 molecules-27-06090-f008:**
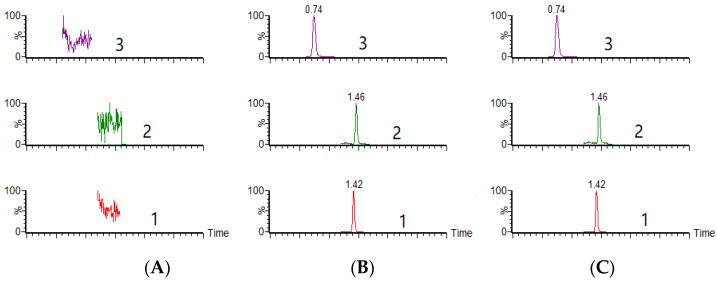
Chromatograms of the analytes and puerarin (IS) in rat plasma: (**A**) blank rat plasma sample; (**B**) blank rat plasma sample spiked with PTD, PTB, and IS based on lower limit of quantification levels; (**C**) rat plasma sample at 30 min after oral administration. (1) puerarin (IS), (2) PTB, (3) PTD.

**Figure 9 molecules-27-06090-f009:**
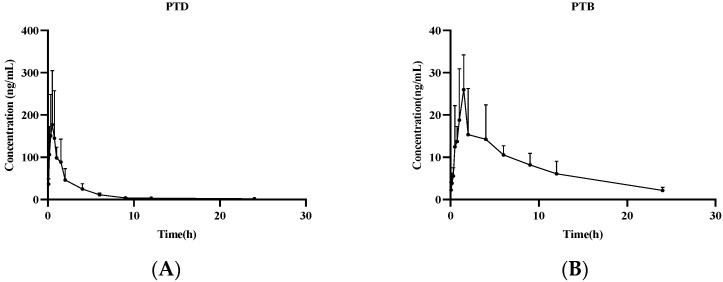
Mean plasma concentration-time curves of PTD (**A**) and PTB (**B**) in rats after oral administration of PTD (27.3 mg/kg).

**Figure 10 molecules-27-06090-f010:**
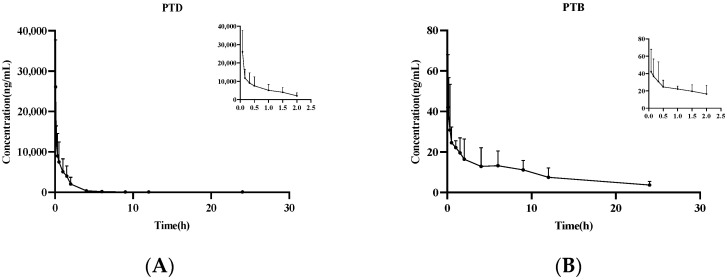
Mean plasma concentration-time curves of PTD (**A**) and PTB (**B**) in rats after intravenous administration of PTD (9.1 mg/kg).

**Table 1 molecules-27-06090-t001:** ^1^H-NMR (CD_3_OD, 600 MHz) and ^13^C-NMR (CD_3_OD, 150 MHz) data of PTD.

Pos.	*δ* _C_	*δ* _H_	HMBC(H→C)
1	132.8/132.1	7.10, 7.10 (d, 8.5)	C-2, 3, 4a, 12a
2	113.1/112.6	6.89, 6.84 (d, 8.5)	C-3, 4, 12b
3	159.3/159.3		
4	110.6/110.0	6.84, 6.78 (d, 2.5)	C-3, 4a, 12b
4a	160.4/159.5		
6a	76.9/76.8	3.92, 3.58 (d, 11.8, 12.3)	C-4a, 7, 8
6b	76.9/76.8	4.42, 4.19 (m)	C-4a, 7, 8
7	73.2/72.9		
7-CHa	68.4/65.8	3.56, 3.41 (m)	C-6, 7, 8
7-CHb	68.4/65.8	3.56, 3.49 (m)	C-6, 7, 8
8-CHa	42.6/40.0	2.56, 2.52 (d, 13.6)	C-6, 7, 9, 8a, 12a
8-CHb	42.6/40.0	2.72, 2.67 (d, 13.6)	C-6, 7, 9, 8a, 12a
8a	128.3/127.9		
9	120.1/119.1	6.75, 6.75 (m)	C-8, 8a, 11
10	145.1/145.1		
11	145.2/145.2		
12 13	117.9/117.6	6.74, 6.69 (s)	C-8a, 10, 12a, 12b
12a	132.1/131.6		
12b	127.6/127.0		
1′	102.2/102.2	4.96, 4.94 (d, 8.1, 7.6)	C-3
2′	74.9/74.9		
3′	78.1/78.1		
4′	71.4/71.4		
5′	78.0/78.0		
6′	62.6/62.5		

**Table 2 molecules-27-06090-t002:** Mass spectral data of PTD (M0) and its metabolites (M1-M4).

	Retention Time (min)	Formula	Calculated (*m*/*z*)	Measured (*m*/*z*)	Error (ppm)	Identification
M0	4.32	C_22_H_26_O_11_	465.1991	465.1399	2.1	Parent
M1	4.55	C_16_H_16_O_6_	303.0863	303.0874	3.1	desugarization
M2	4.33	C_16_H_15_0_5_	288.0992	287.0906	−3.4	Dealkylation, Dehydration
M3	4.60	C_16_H_13_O_5_	286.0836	285.0748	−2.5	Dealkylation, Dehydration, Desaturation
M4	4.49	C_23_H_27_O_12_	496.1575	495.1484	−2.6	Desaturation, Oxidation, Methylation

**Table 3 molecules-27-06090-t003:** Mean values of regression equation of the two compounds.

Administration Mode	Analytes	Linear Regression Equation	*R* ^2^	Linear Range (ng/mL)	LLOQ (ng/mL)
oral	PTD	*Y* = 0.0033*X* − 0.0017	0.9992	1.002~250.50	1.002
	PTB	*Y* = 0.004*X* + 0.0131	0.9981	1.004~100.40	1.004
intravenous	PTD	*Y* = 0.0054*X* − 0.3194	0.9998	1.002~30,060	1.002
	PTB	*Y* = 0.0061*X* + 0.0233	0.9958	1.004~502	1.004

LLOQ: lower limit of quantification.

**Table 4 molecules-27-06090-t004:** Summary of precision and accuracy of the two analytes in rat plasma (*n* = 5).

Analytes	Spiked Concentration (ng/mL)	Intra-Day		Inter-Day	
		Precision RSD (%)	Accuracy (%)	Precision RSD (%)	Accuracy (%)
PTD	5.01	6.69	95.41	7.48	109.38
	250.5	12.82	89.73	8.94	87.26
	15,030	7.15	97.12	3.37	100.64
PTB	2.51	2.78	86.06	4.02	108.76
	20.08	2.33	100.25	6.88	94.12
	251	4.37	94.13	3.48	104.09

RSD: relative standard deviation.

**Table 5 molecules-27-06090-t005:** Summary of recovery and matrix effect of the two analytes in rat plasma (*n* = 5).

Analytes	Spiked Concentration (ng/mL)	Extraction Recovery		Matrix Effect	
		Mean ± SD (%)	RSD (%)	Mean ± SD (%)	RSD (%)
PTD	5.01	92.69 ± 6.78	7.31	95.81 ± 12.10	12.62
	250.5	103.25 ± 4.90	4.74	97.20 ± 9.05	9.31
	15,030	92.84 ± 11.53	12.42	88.83 ± 6.04	6.80
PTB	2.51	91.03 ± 11.03	12.12	96.13 ± 6.11	6.35
	20.08	87.65 ± 4.24	10.18	105.08 ± 8.33	4.84
	251	97.74 ± 9.16	9.37	102.43 ± 12.85	12.55

Mean ± SD: Mean ± standard deviation.

**Table 6 molecules-27-06090-t006:** Stability of the two analytes in rat plasma under various storage conditions (*n* = 5).

Analytes	Conditions	Nominal Concentration (ng/mL)	Measured Concentration (ng/mL) (Mean ± SD)	RSD (%)
PTD	Room temperature for 24 h	5.01	5.61 ± 0.32	5.70
		250.5	241.97 ± 8.73	3.61
		15,030	15,135.44 ± 1324.59	8.75
	4 °C for 24 h	5.01	4.96 ± 0.65	13.1
		250.5	236.32 ± 13.98	5.92
		15,030	15,122.74 ± 959.18	6.34
	There freeze-thaw cycles	5.01	5.23 ± 0.58	11.09
		250.5	260.48 ± 17.36	10.82
		15,030	14,844.73 ± 346.61	2.33
PTB	Room temperature for 24 h	2.51	2.17 ± 0.13	5.99
		20.08	21.81 ± 1.46	6.69
		251	244.31 ± 8.62	3.53
	4 °C for 24 h	2.51	2.32 ± 0.11	4.74
		20.08	24.62 ± 1.12	4.55
		251	223.94 ± 15.73	7.02
	There freeze-thaw cycles	2.51	2.68 ± 0.03	1.12
		20.08	19.43 ± 1.08	5.56
		251	259.84 ± 23.13	8.90

**Table 7 molecules-27-06090-t007:** Pharmacokinetic parameters of PTD and PTB in rat plasma after intragastric administration of PTD (27.3 mg/kg) and intravenous administration of PTD (9.1 mg/kg). Data was expressed as mean ± SD (*n* = 5).

Parameters	Intragastric		Intravenous	
	PTD	PTB	PTD	PTB
AUC_(0–t)_ (ng/mL·h)	382.17 ± 211.15	179.33 ± 44.95	19,364.32 ± 10,497.34	202.20 ± 59.54
AUC_(0–∞)_ (ng/mL·h)	404.56 ± 224.26	204.06 ± 43.91	19,733.20 ± 10,377.52	229.54 ± 71.87
MRT_(0–t)_ (h)	3.06 ± 0.63	7.41 ± 0.77	1.80 ± 0.50	6.45 ± 2.15
MRT_(0–∞)_ (h)	4.15 ± 1.09	11.09 ± 2.32	2.40 ± 0.83	10.46 ± 3.88
t_1/2z_ (h)	3.47 ± 0.78	7.79 ± 1.59	3.65 ± 1.26	6.78 ± 2.69
T_max_ (h)	0.49 ± 0.15	1.42 ± 0.49	-	-
CL_z/F_ (mL/h/kg)	87.51 ± 45.49	141.71 ± 44.73	0.56 ± 0.36	40.47 ± 15.27
V_z/F_ (mL/kg)	415.65 ± 207.32	1646.32 ± 831.76	3.46 ± 3.36	350.38 ± 69.41
C_max_ (ng/mL)	250.45 ± 175.53	31.06 ± 10.47	-	-

AUC_(0–t)_: area under the concentration–time curve from time zero to the last measurable concentration; AUC_(0–∞)_: area under the concentration–time curve from time zero to infinity; CL_z/F_: apparent volume of distribution; C_max_: maximum plasma concentration; MRT_(0–t)_: mean retention time from time zero to the last measurable concentration; MRT_(0–∞)_: mean retention time from time zero to infinity; t_1/2z_: apparent elimination half-life; Tmax: the time to achieve Cmax; V_z/F_:apparent clearance.

**Table 8 molecules-27-06090-t008:** Mass spectra properties of the analytes and internal standards.

Analytes	Precursorion (*m*/*z*)	Production (*m*/*z*)	Cone (V)	Collision (V)	ESI
Puerarin (IS)	417.00	267.00	50	30	+
PTD	465.13	303.08	35	20	+
PTB	303.30	231.10	30	20	−

## Data Availability

The authors declare that all data supporting the findings of this study are available within the article.

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
