# Peer review of "Identification of Protosappanoside D from Caesalpinia decapetala and Evaluation of Its Pharmacokinetic, Metabolism and Pharmacological Activity"

_molecules, 2022, doi:10.3390/molecules27186090_

Round 1
Reviewer 1 Report
The paper entitled “Structural identification, pharmacokinetics and pharmacological activity of Protosappanoside D, a new compound isolated 3 from caesalpinia decapetala” reports isolation of dibenzoxocin derivative named Protosappanoside D from Caesalpinia decapetala, and evaluation i its anti-inflammatory activity. After reviewing, my comments concerning the manuscript are general positive, but before acceptance for publication in Molecules, the results need some modification considering following suggestions:
1. The title of the article is not written according to the rules adopted by Molecules.
2. Isolated protosappanoside D is a component of Caesalpinia decapetala, but there is no information in what amount this compound occurs in the tested 80% ethanol extract. Please provide information on this subject. This information is important, because the determined content of the tested compound in the extract also determines the directions and strength of the activity of the tested extract.
3. Please complete the information from which raw material of C. decapeta the isolation of the compound Protosappanoside D was carried out.
Author Response
Response to Reviewer 1 Comments
Point 1: The title of the article is not written according to the rules adopted by Molecules.
Response 1: Thank you for your comments! It’s deeply sorry for our negligence. According to your suggestion, we have modified its, Please refer to the revised draft for specific modifications (The details are also as follows)
“Structural Identification of Protosappanoside D from Caesalpinia decapetala and Evaluation of Pharmacokinetics, metabolism and Pharmacological Activity”
Point 2: Isolated protosappanoside D is a component of Caesalpinia decapetala, but there is no information in what amount this compound occurs in the tested 80% ethanol extract. Please provide information on this subject. This information is important, because the determined content of the tested compound in the extract also determines the directions and strength of the activity of the tested extract.
Response 2: Thank you for your comments! Accurately weigh 0.1 g of the extract of Caesalpinia decapetala, add 80% ethanol to volume to 25 mL, and this was followed by sonication in an ultrasonic machine for 10 min, and centrifugation for 10 min. After filter and take an appropriate amount of follow-up filtrate and dilute it 20 times with 80% methanol. Then pass through 0.22 μm microporous membrane, and the filtrate was injected into the UPLC-MS/MS system for analysis. The injection information is as follows:
|
Analytes |
Linear regression equation |
Linear range (μg/ml) |
R2 |
peak area (PTD) |
|
Caesalpinia decapetala |
y=180528×+14969 |
0.0006~10 |
0.9971 |
342055 |
Therefore, the amount of PTD in 1g of Caesalpinia decapetala extract was:
(342055-14969)/180528×20×25÷0.1÷1000000×100=0.906%
Point 3: Please complete the information from which raw material of C. decapeta the isolation of the compound Protosappanoside D was carried out.
Response 3: Thank you for your suggestions! It’s deeply sorry for our negligence. According to your suggestion, we have complete the information from which raw material of C. decapeta and was shown in pages 14. Please refer to the revised draft for specific modifications (The details are also as follows)
Caesalpinia decapeta was purchased from the medicinal materials market Wandongqiao, Guiyang City, Guizhou Province, China (2019, harvest), and it was identified as the dried root of Caesalpinia decapetala (Roxb.) Alston by Associate Professor Chunhua Liu (School of Pharmacy, Guizhou Medical University).

Reviewer 2 Report
The manuscript "Structural identification, pharmacokinetics and pharmacologi-2 cal activity of Protosappanoside D, a new compound isolated 3 from caesalpinia decapetala" is well designed, executed and written. It reports a new compound Protosappanoside D (PTD) that is studied comprehensively. There are some minor concerns related to this manuscript, which are as below.
1. The natural products previously identified in Caesalpinia decapetala should be discussed in the introduction part along with any pharmacological activity reported so far.
2. The authors are advised to refine the language, as there are some minor grammatical errors in the manuscript.
Author Response
Response to Reviewer 2 Comments
Point 1: The natural products previously identified in Caesalpinia decapetala should be discussed in the introduction part along with any pharmacological activity reported so far.
Response 1: Thank you for your reminder! According to your suggestion, the content of the article has been modified and references have been added was shown in pages 1. Please refer to the revised draft for specific modifications. (The details are also as follows)
Caesalpinia decapeta (C. decapetala) [Latin name:Caesalpinia decapetala (Roth) Alston] is a traditional medicinal plant of the Miao people. It belongs in the Fabaceae family and has been used to treat colds, fever, rheumatic pain, etc for many years. In previous studies, it was found that the extract of C. decapetala mainly contains compounds such as phenolics, flavonoids, and terpenoids, which has good anti-inflammatory, antiviral and antitumor properties. For instance, gallic acid has a good proapoptotic effect on A-489 renal cancer cells, and can also inhibit the secretion of inflammatory factors induced by LPS; sappanchalcones can significantly reduce the secretion of tumour necrosis factor alpha (TNF-α), interleukin-1β (IL-1β) and interleukin-6 (IL-6) to improve the joint destruction and surface erosion claws of CIA mice; luteolin can play an anti-inflammatory role by inhibiting NF-κB pathway, and inhibiting STAT3 pathway showing a strong ability to inhibit liver fibrosis; protosappanin B (PTB) has obvious pharmacological activities such as antitumour, antioxidant and anti-inflammatory activities.
Point 2: The authors are advised to refine the language, as there are some minor grammatical errors in the manuscript.
Response 2: Thanks for your suggestions! We are sorry for some errors in English and grammar. This manuscript has checked by a native English-speaking colleague, and have made some corrections to it.
Reviewer 3 Report
Comments:
1. In cell viability test (Figure 3A), it seems there is no significant difference between 300 μmol/L and 400 μmol/L concentration of PTD on cell viability and at 500 μmol/L concentration, PTD impose significant effect on cell viability. Can authors explain the observation of this result? Also, in Figure 3B, authors have not performed the test at 500 μmol/L concentration of PTB, any logical explanation?
2. In Figure 4 (effect of PTD and PTB on the secretion of IL-6, TNF-α, IL-1β, and NO in RAW264.7 cells), there are few of issues. First, statistical significance values have not been shown with comparable lines between the bars (between model and treated). Is the significance value shown on model compared with the control (**)? In IL1β result, it seems the PTD and PTB has significant difference compared to the model, but the authors showed that both are equally statistically significant (###), can authors explain this? In NO result, it seems PTD has no effect on NO secretion, why?
3. Did authors test the effect of PTD and PTB on the secretion of IL-6, TNF-α, IL-1β, and NO using other cell lines?
4. In the pharmacokinetic study, is the PTD dose administered to rats is nominal or well standard?
5. Did authors carried out the pharmacokinetic study using PTB also?
6. How was the bioavailability score (percentage) calculated?
7. The quality of English is poor. There are several grammatical errors in the manuscript and those need to be corrected.
8. Typo error: line 157, Materia? Metabolites message?
Author Response
Response to Reviewer 3 Comments
Point 1: In cell viability test (Figure 3A), it seems there is no significant difference between 300 μmol/L and 400 μmol/L concentration of PTD on cell viability and at 500 μmol/L concentration, PTD impose significant effect on cell viability. Can authors explain the observation of this result? Also, in Figure 3B, authors have not performed the test at 500 μmol/L concentration of PTB, any logical explanation?
Response 1: Thank you for your questions! We have answered your questions one by one, as follows:
(1)The effect of drugs on cell growth is concentration dependent. When the drug concentration reaches 300 μmol/L and 400 μmol/L has a significant inhibitory effect on cell growth, so 500 μmol/L had stronger inhibitory effect on cell growth, and the effect was more significant. After carefully examining the data of parallel experiments, it was found that 300 μmol/L and 400 μmol/L had little effect on cells (as shown in Figure 1 below:), but 300 μmol/L does have a slightly smaller effect on cells. However, there was no difference between the 300 μmol/L and 400 μmol/L on cells was showed in Figure 3A, which may be caused by experimental errors. (It has been modified as the data of parallel experiment 2 in the text)
(A) (B)
Figure 1. Cytotoxic effects of PTD on RAW264.7 cells. (A) and (B) represent the data of the first parallel experiment and the second parallel experiment, respectively.
(2)Drugs may be toxic to cells. Before investigating the anti-inflammatory effects of PTD and PTB, the drug concentration should be screened for safety to select a concentration that has no effect on cell growth for the experiment. The concentration range of PTD and PTB in this study is only set for cytotoxicity investigation. The results show that the safe concentration of PTD was≤200 μmol/L, the safe concentration of PTB was≤300 μmol/L. The activity experiment was conducted with 200 μmol/L, the concentration is far less than 500 μmol/L.
Point 2: In Figure 4 (effect of PTD and PTB on the secretion of IL-6, TNF-α, IL-1β, and NO in RAW264.7 cells), there are few of issues. First, statistical significance values have not been shown with comparable lines between the bars (between model and treated). Is the significance value shown on model compared with the control (**)? In IL1β result, it seems the PTD and PTB has significant difference compared to the model, but the authors showed that both are equally statistically significant (###), can authors explain this? In NO result, it seems PTD has no effect on NO secretion, why?
Response 2: Thank you for your questions! We have answered your questions one by one, as follows:
(1) The “*” symbol in the figure indicates the statistical significance (p: pvalue) of the blank group compared with the model group. When comparing the control group with the model group, 0.01<p<0.05 is represented by “*”, 0.001<p< 0.01 is represented by “**”, and P<0.001 is represented by “***”; similarly, the “#” symbol indicates the statistical significance of the comparison between the treated group and the model group. When comparing the treated group with the model group, 0.01<p<0.05 is represented by “#”, 0.001 < p < 0.01 is represented by “##” and P < 0.001 is represented by “###”.
(2) When p<0.001, represented by “###”. In IL-1β result, comparing with the model group, PTD group could significantly down the release of IL-1β (p<0.001), and PTB group could also significantly down he release of IL-1β (p<0.001). So, showed that both are equally statistically significant (###) to represent the PTD and PTB group compared to the model group.
(3) NF-κB, MAPK and PI3K/Akt signaling pathways participate in the regulation, forming a complex mechanism network during inflammation. Free radical NO is produced by inducible nitric oxide synthase (iNOS) and mediates many biological functions. It is well established that iNOS expression by LPS in macrophages is regulated by the transcription factors nuclear factor-κB (NF-κB) and AP-1. PTD can’t down regulate the secretion of no, and its anti-inflammatory effect may be that it can’t inhibit the expression of iNOS regulated by NF- κB and AP-1. This speculation needs further experiments to confirm.
Point 3: Did authors test the effect of PTD and PTB on the secretion of IL-6, TNF-α, IL-1β, and NO using other cell lines?
Response 3: Thank you for your comments! Macrophages play an important role in the inflammatory response. They are the central cells that initiate the production of inflammatory mediators and inflammatory factors in vivo, and also the main cells that regulate the inflammatory response. RAW264.7 cells is considered as one of the best models of macrophages, and is the most commonly used in vitro research model for screening anti-inflammatory agents and studying inflammation. LPS is a microbial product from the cell wall of bacteria (gram-negative), generally bonds to toll-like receptor 4 (TLR-4) in macrophages that ultimately the NF-κB and Akt signaling pathways excessive activation and generates many pro-inflammatory mediators, including NO, IL-1β, IL-6 and TNF-α, thereby triggering inflammation. Therefore, only the classical cell line RAW264.7 was selected for the anti-inflammatory activity in this study.
Point 4: In the pharmacokinetic study, is the PTD dose administered to rats is nominal or well standard?
Response 4: Thank you for your questions! In the pharmacokinetic experiment, the dose of PTD was referred to the effective dose in the pharmacodynamic experiment of Caesalpinia decapeta extract (This part of the research will be published). According to the provincial standard, the dosage of Caesalpinia decapeta in rats is 1.575 g/kg (the crude herb), and the extraction rate of the extract was 10%. So, the dosage of Caesalpinia decapeta extract in rats is 0.1575 g/kg. However, in the pharmacodynamic experiment in the extracted object, the dose of 3.3 g/kg was administered to rats, which was about 20 times of the provincial standard. The content of PTD in the extract is about 0.906%. According to the local standard, the dosage of PTD is 1.43 mg/kg, and the dosage of this experiment is 27.3 mg/kg, which is also about 20 times of the provincial standard.
Point 5: Did authors carried out the pharmacokinetic study using PTB also?
Response 5: Thank you for your questions! This study aim to explore the dynamic and transformation processes of PTD in vivo. However, prior studies had found that PTB and PTD in the extract of C. decapetala were able to enter the bloodstream. To eliminate any interference of PTB in the extract, only PTD was used in the metabolism and pharmacokinetics studies in vivo. In the metabolic study found that PTD can be converted to PTB after removing its glycosidic bond. In pharmacokinetic experiments, it was found that PTD can be converted into PTB in vivo after oral and intravenous injection. The above experiments confirmed that PTD can be transformed into PTB in vivo.
Point 6: How was the bioavailability score (percentage) calculated?
Response 6: Thank you for your questions! Bioavailability is calculated as follows:
Bioavailability= (AUCPO×Div)/(AUCiv×Dpo)×100% (po: Oral administration, iv: Intravenous injection, D: dose of administration, AUCpo=382.17, Div=9.1, AUCiv=19364.32, Dpo=27.3). So, the bioavailability of PTD is: ( 382.17x9.1)/(19364.32x27.3)x100%=0.65%.
Point 7: The quality of English is poor. There are several grammatical errors in the manuscript and those need to be corrected.
Response 7: Thanks for your suggestions! We are sorry for some errors in English and grammar. This manuscript has checked by a native English-speaking colleague, and have made some corrections to it.
Point 8: Typo error: line 157, Materia? Metabolites message?
Response 8: Thanks for your kind reminder! It’s sorry for the typing error caused by our omissions in writing. According to your suggestions, we have revised it and checked the words in the article one by one. Please refer to the revised draft for specific modifications.

Round 2
Reviewer 1 Report
The Authors revised the manuscript as suggested by the reviewer. Therefore, I recommend this manuscript to be published in Molecules.
Author Response
Dear Reviewer,
On behalf of my co-authors, we thank you very much for giving us an opportunity to revise our manuscript, and we also appreciate you very much for your constructive suggestions on our manuscript entitled “Identification of Protosappanoside D from Caesalpinia decapetala and Evaluation of its Pharmacokinetic, Metabolism and Pharmacological Activity” (Manuscript ID: molecules-1902320 ).
Thank you and best regards.
Yours sincerely,
Wensha Meng
Gui Zhou Medical University
Email: 2684921096@qq.com